# The Pterygomandibular Space: A Volumetric Evaluation Using the Novel A-Silicone Injections Method

**DOI:** 10.3390/diagnostics14111161

**Published:** 2024-05-31

**Authors:** Hadi Darawsheh, Ali Alsaegh, Elena Kanukoeva, Rinat Saleev, Gulshat Saleeva, Beatrice Volel, Natalia Kireeva, Ekaterina Rebrova, Yuriy L. Vasil’ev

**Affiliations:** 1N.V. Sklifosovskiy Institute of Clinical Medicine, I.M. Sechenov First Moscow State Medical University (Sechenov University), 119435 Moscow, Russia; hadi.darawsheh@gmail.com (H.D.); beatrice.volel@gmail.com (B.V.); kireeva_n_v_1@staff.sechenov.ru (N.K.); rebrova_e_v@staff.sechenov.ru (E.R.); 2Medical Institute, Patrice Lumumba Peoples’ Friendship University, 117198 Moscow, Russia; ali0alsaegh@gmail.com; 3Dental Department, Russian State Social University, 129226 Moscow, Russia; elena.kanukoewa@yandex.ru; 4Dental Department, Kazan State Medical University, 420012 Kazan, Russia; rinat.saleev@gmail.com (R.S.); rin-gul@mail.ru (G.S.)

**Keywords:** anatomy, anesthesia, dentistry, local anesthesia, mandibular nerve, pterygomandibular space

## Abstract

Inferior alveolar nerve block (IANB) is one of the most common procedures in operative dentistry, and a deep understanding of the normal anatomical variation of the pterygomandibular space (PM) is essential for its safe and successful administration. This cadaveric anatomical study aimed to use A-silicone injections to evaluate the volume of the PM. This study was conducted using 46 human cadaver heads (25 males and 21 females). A craniometric analysis was performed using the cadavers’ ages, the number of silicone cartridges (carpules) used to fill the pterygomandibular space, Izard’s Facial Index (FI), and the Cranial Index (CI). A Halstead mandibular block was performed by injecting 1.7 mL A-silicone cartridges (as an equivalent to standard local aesthetic carpules volume) into the PM. The cured silicone was extracted from the dissected mandibles. The volume (length, width, and thickness) of the extracted silicone and the number of silicone cartridges used to fill the space were evaluated. The results showed that there are statistically significant positive correlations between the CI and the width of the right PM, as well as the width and length of the left PM. A statistically significant correlation was found between the width of the left PM and the age of the cadaveric heads; the higher the age, the thicker the space on both sides. The volume of the PM corresponded to 1.5 cartridges on average.

## 1. Introduction

Effective pain management is fundamental for quality dental care. It is essential to achieve maximum pain control in the shortest amount of time with the least amount of discomfort for the patient. The mandible is difficult to anesthetize in comparison to the maxilla because the outer layer of the mandibular cortical bone is thick and non-porous. Various techniques are available for local mandibular anesthesia and include the traditional inferior alveolar nerve block (IANB), the Akinosi–Vazirani method, and the Gow-Gates technique [1,2]. However, each technique has advantages and disadvantages [2,3,4].

IANB is the most common procedure in operative dentistry [5,6,7]. Technically, the needle should be positioned in closest proximity to the inferior alveolar nerve. To maximize effectiveness, it is advised to inject enough local anesthetic into the pterygomandibular space (PM). Studies suggest that this technique prevents nerve injury [8].

In some cases, it is challenging to achieve sufficient nerve block effectiveness because of the anatomical variability of the PM, which requires the administration of a higher-than-average dose of anesthetic, which can lead to undesirable phenomena [9]. The PM is filled by connective tissue and formed by the muscles, and it also includes the inferior alveolar nerve, artery, and vein. Many published studies have used computer tomography to determine the volume and morphology of the PM [10,11]. However, according to the best of our knowledge, in the English literature, no previous studies have determined the volume of the PM using the A-silicone injection method. Therefore, the goal of the research is to evaluate the volume of the PM using the A-silicone injections method. The proposed null hypothesis in this study is that there is no correlation between craniometric parameters (the Cranial Index and Izard’s Facial Index) and the volume of PM (length, width, and thickness) on the left and right.

## 2. Materials and Methods

The present study was conducted at the Department of Operative Surgery and Topographic Anatomy, N.V. Sklifosovskiy Institute of Clinical Medicine, Sechenov University, Moscow, Russia. The sample mean included forty-six human cadaver heads (25 males and 21 females). The age of the heads ranged from 53 to 87 years (Average age = 72.5 years). The source of the cadaveric specimens was the Skolkovo Institute of Anatomy, Moscow, Russia.

A craniometric analysis was performed using the Cranial Index (CI) and Izard’s Facial Index (FI). The first index was calculated using the following mathematical formula: CI = (the transverse diameter × 100)/the longitudinal diameter. The transverse diameter is the distances between the parietal eminences (eu-eu). The longitudinal diameter is the distance from the glabella (g) to the external occipital protuberance (ops). The collected data were split up into three groups (skull shapes): doichocrania = up to 74.9%, mesocrania 75–79.9%, and brachycrania = 80% or higher.

The second index was calculated using the following mathematical formula: FI = the facial length × 100)/the facial width. The facial length was measured from the ophrion point (oph) to the gnathion point (gn). The ophrion point (oph) is located at the intersection of the midline of the face and the tangent to the brow ridges. The gnathion point (gn) is on the midline of the face, under the chin. Th facial width was determined between the most prominent points on the zygomatic arches (zy-zy). [5,12]. The obtained data were divided into three groups (facial types): broad (wide) facial type = 96% or less, medium facial type = 97–103%, and narrow facial type = 104% or higher.

A Halstead nerve block was performed by injecting 1.7 mL A-Silicone cartridges (as equivalent to standard local aesthetic carpules volume) into the PM [4]. The injected material (S4 Suhy, Bisico^®^, Bielefeld, Germany) is an additional curing, extremely light-bodied correction dental impression material with super-hydrophilic properties. The volume of the cadaveric PM was evaluated as follows: after the injected silicone was fully cured, it was carefully extracted by sawing the mandible, preserving all the contours and the wholeness of the extracted material (silicone). Exclusion criteria: (1) material (cured silicone) obtained by noncompliance with the correct technique of mandibular anesthesia; (2) any violation of the integrity of the cured silicone during its extraction.

Measurements were carried out using a digital caliper (country of origin: Taiwan). The measurement range was 0–15 cm/0–6′ and the error rate was 0.01 mm/0.0005′. Volumetric evaluation (length, width, and thickness) of the sample mean was carried out, and the number of cartridges used to fill in the PM was calculated (Figure 1).

Statistical analysis: metric data were presented as mean ± standard deviation. Multiple comparisons were performed using non-parametric tests (Mann–Whitney and Kruskal–Wallis). A *p*-value less than 0.05 was considered significant.

The study was conducted according to the ethical principles of the Declaration of Helsinki and was approved by the Ethics Committee of Sechenov University, Moscow, Russia (Number 02-23; 26 January 2023).

## 3. Results

The data presented in Table 1 show the results of the analysis of the volume of the PM using the silicone injection method.

The median FI was 105.13%, and the median CI was 58.51%. The PM was characterized by the following ratio of length and width: 3.6 and 2.2 on the right PM; and 3.6 and 2.17 on the left PM. A broad (wide) facial type was detected in 13.04% of the examined heads and a medium facial type was detected in 17.39%. Narrow facial type prevailed and accounted for 69.56% of heads. Dolichocrania prevailed (82.61%), while mesocrania and brachycrania occurred in equal proportions (8.7%, respectively).

The study of the relationship between the length and width of the PM with the FI was carried out in two ways: graphically using the first group of scatterplots (Figure 2) and by performing correlation analysis. The scatterplots show a relatively uniform distribution of values, without defined dependencies. The obtained data were correlated using Spearman’s rank correlation coefficient (ρ): (1) The FI and the length of the right PM showed a very weak correlation (r = 0.080, *p* = 0.598). (2) The FI and the width of the right PM showed a very weak correlation (r = −0.115, *p* = 0.443). (3) The FI and the length of the left PM showed a very weak correlation (r = −0.158, *p* = 0.293). (4) The FI and width of the left PM showed no correlation (r = −0.051, *p* = 0.734).

Based on the results of the correlation analysis, it is difficult to draw an unambiguous conclusion about the presence of statistically significant relationships between the length and width of the PM and the FI, regardless of the side on which the measurements were made. The *p*-value in all cases exceeded 0.05, indicating that there is insufficient evidence to reject the null hypothesis of no correlation. However, there were only a few very weak multi-directional correlations.

The first group of box-plot diagrams shows the distribution of the values of the length (a) and width (b) of the right PM based on the facial type (Figure 3). The first diagram (a) shows no significant differences in the length of the space on the right side, regardless of the facial type, which is confirmed statistically (the Kruskal–Wallis test (χ^2^ = 0.194, degrees of freedom = 2, *p* = 0.907)). The second diagram (b) shows that, in the narrow facial type, the width values on the right side are slightly higher compared to the other facial types (the Kruskal–Wallis test (χ^2^ = 0.1058, degrees of freedom = 2, *p* = 0.948)).

The second group of box-plot diagrams (Figure 4) demonstrates how the distribution of the values of the length (a) and width (b) of the left PM varies based on the facial type. The first diagram (a) shows that, in the median, the length values on the left PM for the narrow facial type are slightly smaller compared to other facial types. Simultaneously, no statistically significant differences were found in the Kruskal–Wallis test (χ^2^ = 1.587, degrees of freedom = 2, *p* = 0.452). The second diagram (b) shows that the width values on the left PM for the broad (wide) facial type, on average, exceeded those for the other two facial types, but without statistically significant differences in the Kruskal–Wallis test (χ^2^ = 1.288, degrees of freedom = 2, *p* = 0.525).

The second group of scatterplots shows the relationship between the length and width of the PM with the CI (Figure 5). The obtained data were correlated using Spearman’s rank correlation coefficient (ρ): (1) The CI and the length on the right side showed a very weak correlation (r = 0.209, *p* = 0.163). (2) The CI and the width on the right side showed a very weak correlation (r = 0.286, *p* = 0.054). (3) The CI and length on the left side showed a weak correlation (r = 0.366 (95% CI from 0.085 to 0.609) *p* = 0.013). (4) The CI and the width on the left side showed a weak correlation (r = 0.359 (95% CI from 0.059 to 0.610) *p* = 0.014). Results of the correlation analysis showed statistically significant positive correlations of the CI with the width of the right PM (borderline significance *p* = 0.054), as well as the length and width on the left PM (*p* = 0.013 and *p* = 0.014, respectively).

The third group of box-plot diagrams shows the distribution of the length (a) and width (b) of the PM on the right PM at different gradations of the CI (Figure 6). On average, the length values of the left PM (a) were higher in mesocrania, which is confirmed statistically (the Kruskal–Wallis test (χ^2^ = 8.7358, degrees of freedom = 2, *p* = 0.013). Considering multiple comparisons, the significance of differences was revealed between mesocrania and dolichocrania (*p* = 0.019), as well as mesocrania and brachycrania (*p* = 0.043). Similar results were observed for the width values of the right PM (b), but without statistically significant differences (the Kruskal–Wallis test (χ^2^ = 4.3022, degrees of freedom = 2, *p* = 0.116).

The fourth group of box-plot diagrams shows the distribution of the length (A) and width (B) of the PM in different categories (groups) of the CI (Figure 7). The length values of the left PM (A) were slightly higher in mesocrania (the Kruskal–Wallis test (χ^2^ = 2.2862, degrees of freedom = 2, *p* = 0.319)). On average, the width of the left PM (B) was lower in dolichocrania (the Kruskal–Wallis test (χ^2^ = 5.6186, degrees of freedom = 2, *p* = 0.06)).

The correlation between the thickness of the PM and the FI was studied (Figure 8). Both a visual evaluation (third group of scatter plots) and correlation analysis were carried out. The Spearman’s correlation coefficient (ρ) showed a weak positive correlation, with statistical significance at the level of 0.1: r = 0.275, *p* = 0.064.

The fifth group of box-plot diagrams (Figure 9) shows the differences in the thickness of the left PM (a) and right PM (b), considering the FI. In the medium facial type, the thickness of the space from the nerve entry point on the left PM (a) was slightly higher than in other facial types (the Kruskal–Wallis test (χ^2^ = 1.7101, degrees of freedom = 2, *p* = 0.425)). On the right PM (b), greater thickness values were observed in the narrow facial type (the Kruskal–Wallis test (χ^2^ = 1.1498, degrees of freedom = 2, *p* = 0.563)).

The results of the study of the relationship between the thickness of the PM and the CI are presented as a visual assessment of scatterplots (fourth group) and correlation analysis. The scatterplots showed a slight increase in thickness with an increase in the CI (Figure 10). The Spearman correlation coefficient was +0.166, *p* = 0.2692, which showed a statistically insignificant, very weak, positive relationship.

The sixth group of box-plot diagrams shows the distribution of PM thickness values on the left PM (A) and right PM (B), considering the CI (Figure 11). The thickness of the space on the left PM (A) was slightly higher in mesocrania, without statistically significant differences between the three groups (the Kruskal–Wallis test (χ^2^ = 0.0619, degrees of freedom = 2, *p* = 0.969)). For the thickness of the right PM (B), higher values were observed in dolichocrania and mesocrania, and, significantly, lower values were observed in brachycrania (Kruskal–Wallis test (χ^2^ = 2.1367, degrees of freedom = 2, *p* = 0.344)).

An analysis was carried out on the correlations and size of the PM and the age of the cadavers (below are the correlation coefficient ρ and the value of *p*). The length on the right PM represented a very weak correlation with no statistical significance (r = −0.064; *p* = 0.6735). The width of the right PM represented a very weak correlation with a borderline significance (r = 0.291; *p* = 0.050). The length of the left PM showed a very weak correlation with no statistical significance (r = −0.080; *p* = 0.5981). The width of the left PM showed a statistically significant weak correlation (r = −0.330 (95% CI −0.546 to −0.066); *p* = 0.025). The thickness of the left PM (r = 0.315 (95% CI 0.036 to 0.544); *p* = 0.03304). The thickness of the right PM showed a positive, statistically significant weak correlation (r = 0.338 (95% CI 0.023 to 0.629); *p* = 0.02137).

A statistically significant correlation was found between the width of the left PM and the age of the cadaver (*p* = 0.025); in elderly cadavers, this parameter was lower. In addition, with increasing age, the thickness of the space on both the left and right sides also increased (*p* = 0.278; *p* = 0.061).

The fifth group of scatterplots (Figure 12) shows relationship between the FI and the number of cartridges. The correlation of the number of cartridges on the right PM (ρ = −0.023; *p* = 0.878). The correlation on the left PM (ρ = −0.065; *p* = 0.666). In both cases, no significant correlations were found.

Figure 13 shows the relationship between the CI and the number of cartridges (correlation on the right is *p* = 0.278; *p* = 0.061).

The correlation between the number of cartridges for the left PM and the FI was positive, weak, and statistically significant (r = 0.440, *p* = 0.002, 95% CI = 0.095–0.735).

The sixth group of scatterplots (Figure 13) provides a visual assessment of the relationship between the CI and the number of cartridges. A statistically significant weak positive correlation was found between the CI and the number of cartridges on the left side (*p* = 0.002). On both sides, there is a significant correlation between thickness and the number of cartridges (on the left side (r = 0.323 (95% CI 0.037 to 0.567), *p* = 0.029), and on the right side (r = 0.522 (95% CI 0.260 to 0.716), *p* = 0.0002)).

The seventh group of scatterplots (Figure 14) presents the results of estimating the correlation between the thickness of the nerve entry point and the number of cartridges.

## 4. Discussion

IANB is a routine procedure in operative dentistry. Technically, local anesthetics are administered to the PM, through which the lingual and the inferior alveolar nerves pass. A thorough understanding of the anatomic variation of PM is essential for a safe and successful IANB [13]. The anatomical structure of the PM is variable and complex and must be always considered during local anesthesia, dental treatment, surgical interventions, and diagnostic procedures [14,15].

Historically, the development of cone-beam computed tomography contributed to the observation of the mandibular foramen, the associated canal, and the course of the inferior alveolar nerve [16,17,18]. According to Murphy and Grundy, the volume of a PM is approximately 2.0 mL, and these authors injected 2.0 mL of anesthetic [19]. Gow-Gates and Watson sequentially injected 2.2 mL of the local anesthetic [20]. Of the estimated volume of the PM, 2.2 mL of the dose fills the space and probably spreads in the anterior lower direction. Levy injected 3.0 mL of local anesthetic and achieved a higher success rate (77%) [21]. A higher dose level is an essential factor for successful anesthesia.

To increase the success rate of local anesthesia, it is essential to consider extra-oral landmarks, such as the expansion degrees of the mandibular ramus, the height, and width of the mandibular ramus, as well as intra-oral landmarks [22,23]. In a retrospective study, You et al. (2015) concluded that the failure rate of IANB was substantially greater in retrognathic mandibles than in prognathic mandibles and normal mandibles [24]. According to the authors, this is because the distance between the tip of the condyle and the mandibular foramen is substantially shorter in retrognathia and, as a result, the position of the mandibular foramen is higher than in normal mandibles.

When the needle is inserted above the occlusal plane for performing regular inferior alveolar nerve anesthesia, the anesthetic solution is injected below the mandibular foramen, resulting in a higher failure rate. Moreover, in retrognathic mandibles, an insufficient mouth opening is considered to cause inadequate anesthesia of the inferior alveolar nerve because of the short length of the condyle [25,26]. In different mandibular skeletal shapes (prognathic, retrognathic, and normal mandibles), the position of the mandibular foramen varies, which highlights the importance of implementing methods that do not depend on its position, such as Gow-Gates and Vazirani−Akinosi [27,28].

The published paper of Darawsheh et al. (2024) [29] presented significant results using multi-slice computed tomography to evaluate the volume of PM in various shapes of the mandible. Results of Darawsheh’s et al. (2024) study showed that the volume could vary significantly depending on the anatomical shape of the mandible; the minimum volume in the leptogenic mandibular shape was 1.98 mL and the largest volume was found in the brachygenic mandibular shape (2.05 mL) [29].

According to Leven (1987), a 1.8 mL anesthetic solution fills the PM, regardless of the administration technique. Therefore, the author believes that the term “torusal anesthesia” is non-evidence-based, and accordingly, we believe that implementing it is scientifically unsuitable. Clinically, dentists use a technique called IANB [30,31].

In the literature, there is a single and non-correlated data point on the volume of the PM. For example, according to Madrid et al. (1993) [32], the volume of the PM is superior to the value usually reported in the dental literature. An amount of 4.8 to 5.8 mL of anesthetic solution is sufficient for filling the PM according to denture. We considered that this study is limited to dimensions assessed using unfixed material and by injecting and layer-by-layer dissection. Using materials that are prone to expansion or a temperature regime that differs sharply from the intravital body temperature, a sharp distortion of the boundaries is possible, as well as a decrease or increase in tissue resistance [32].

Studies have suggested several methods for anatomic and volumetric evaluation, including computed tomography and magnetic resonance examination [33,34]. Previous studies in the English literature did not mention the implementation of silicone injections to evaluate the volume of the PM. In the present study, silicone was implemented, which represents a dental impression material that is usually used in day-to-day dental practice [33,34,35,36,37]. The authors believe that this method is realistic for evaluating normal anatomical structures, including, but not limited to the PM. The present study contributes to the body of knowledge by evaluating the volume of the PM and validating the silicone method. However, there is a need for larger studies for definitive conclusions on the validity of this method.

## 5. Conclusions

A statistically significant correlation was found between the width of the left PM and the age of the patient; the higher the age, the thicker the space on both sides (*p* < 0.05). On average, the volume of space corresponded to the volume of 1.5 dental local anesthetic cartridges (2.55 mL).

## Figures and Tables

**Figure 1 diagnostics-14-01161-f001:**
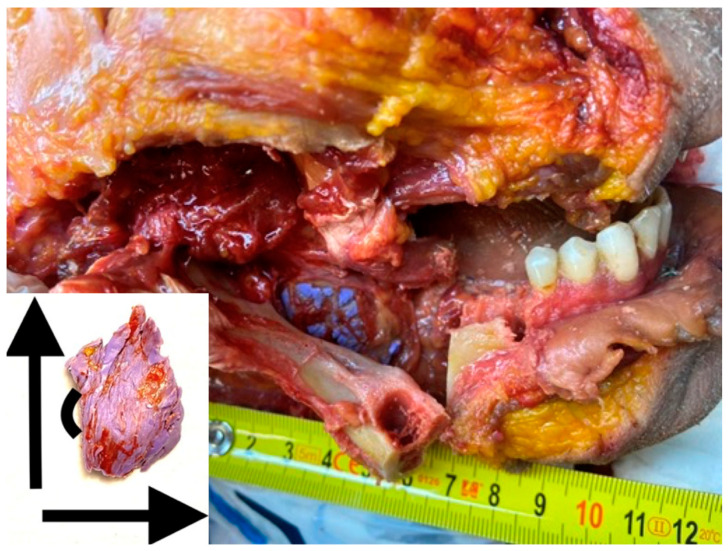
An example of measuring the volume of the PM and the appearance of the cured A-silicone impression with spatial orientation.

**Figure 2 diagnostics-14-01161-f002:**
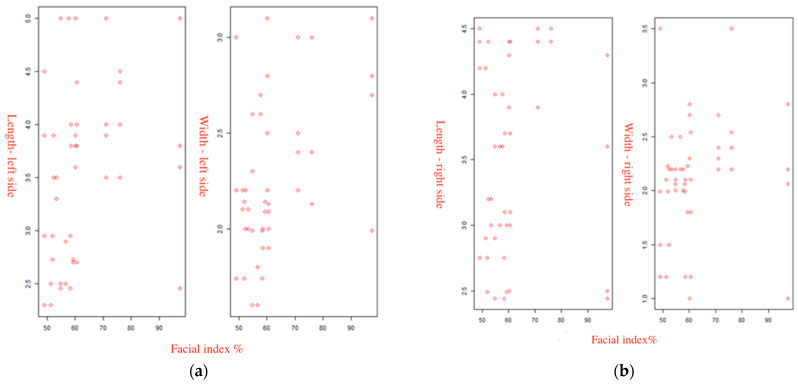
Scatterplots of the relationship of linear dimensions of the PM with the FI. (**a**) The length and width on the left PM. (**b**) The length and width on the right PM.

**Figure 3 diagnostics-14-01161-f003:**
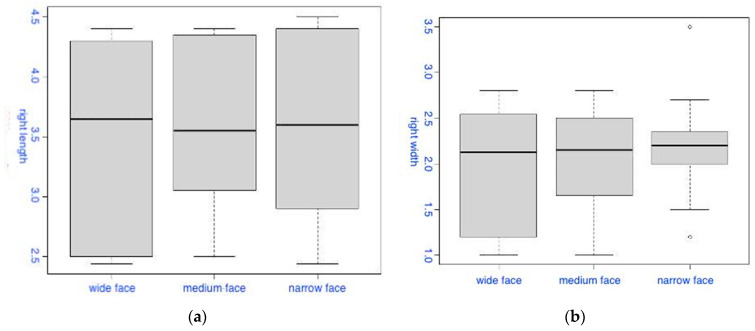
Box-plot diagrams of the distribution of the length (**a**) and width (**b**) of the PM on the right side based on the facial type.

**Figure 4 diagnostics-14-01161-f004:**
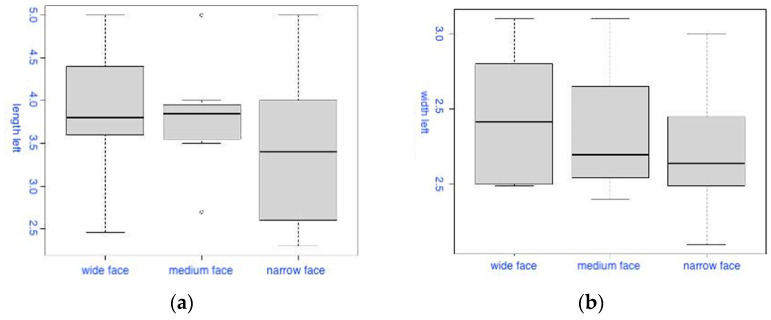
Box-plot diagrams of the distribution of the length (**a**) and width (**b**) of the PM on the left side based on the facial type.

**Figure 5 diagnostics-14-01161-f005:**
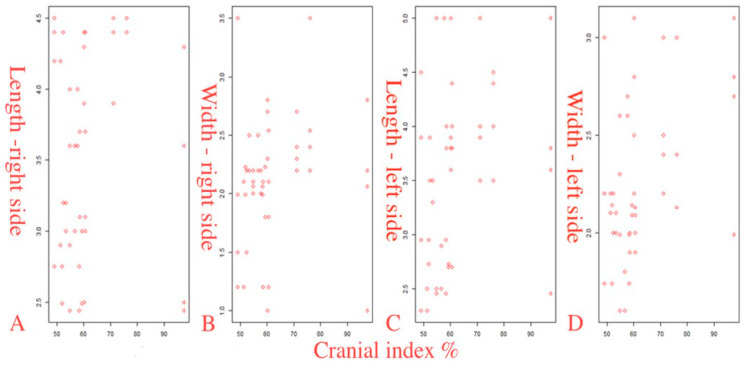
Scatterplots of the relationship of the length and width of the PM with the CI (**A**–**D**).

**Figure 6 diagnostics-14-01161-f006:**
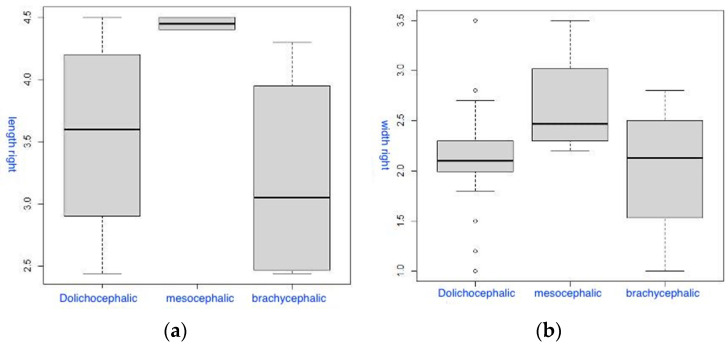
Box-plot diagrams of the distribution of the length (**a**) and width (**b**) of the right PM based on the type of skull.

**Figure 7 diagnostics-14-01161-f007:**
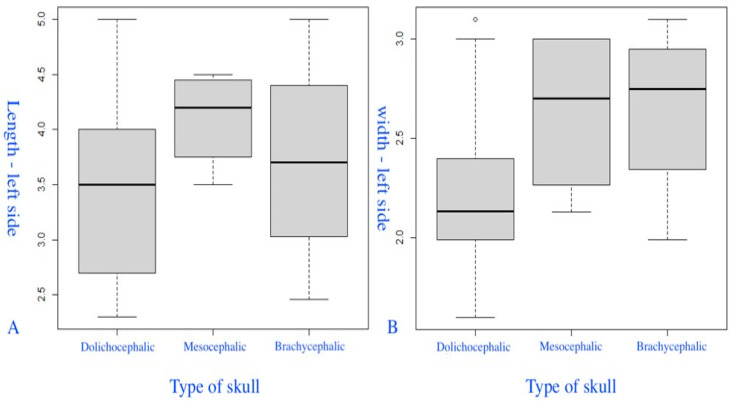
Box-plot diagrams of the distribution of length (**A**) and width (**B**) of the left PM with different types of skull.

**Figure 8 diagnostics-14-01161-f008:**
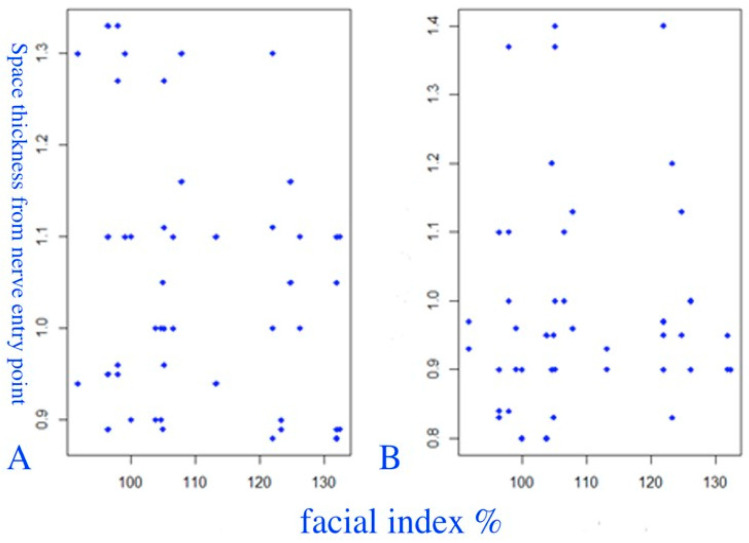
Scatter plots of the relationship between the thickness of the PM and the FI. (**A**) The right PM. (**B**) The left PM.

**Figure 9 diagnostics-14-01161-f009:**
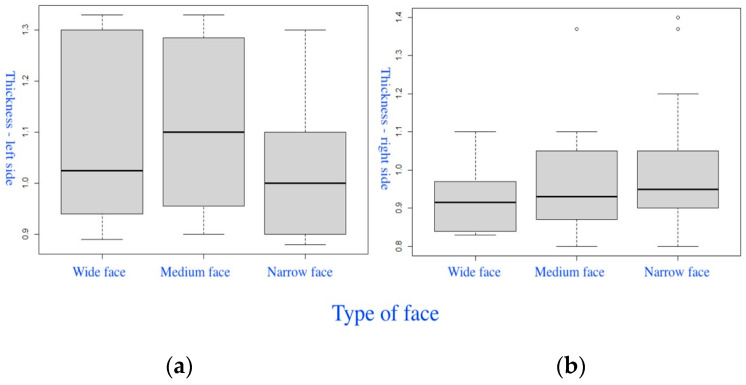
Box-plot diagrams of the distribution of the thickness of the left PM (**a**) and right PM (**b**) in different facial types.

**Figure 10 diagnostics-14-01161-f010:**
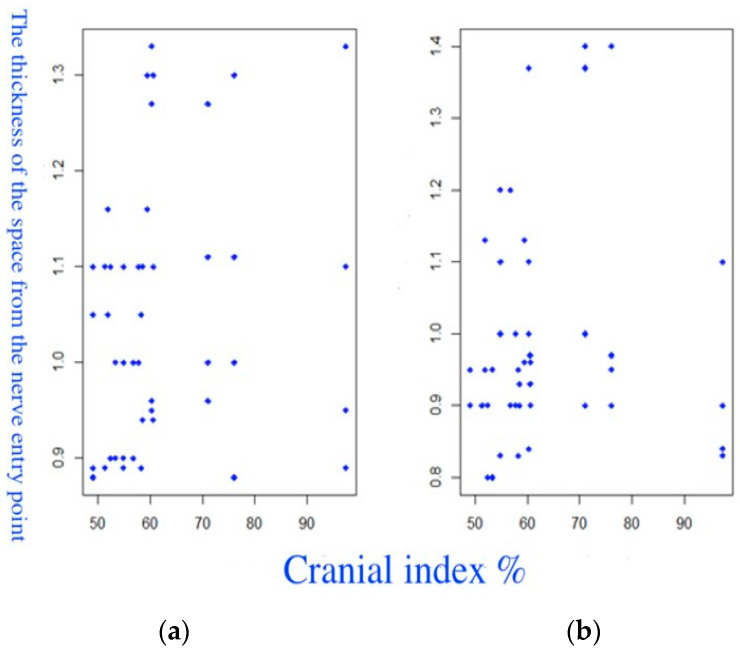
Scatterplots of the relationship between the thickness of the PM and the CI. (**a**) The right PM. (**b**) The left PM.

**Figure 11 diagnostics-14-01161-f011:**
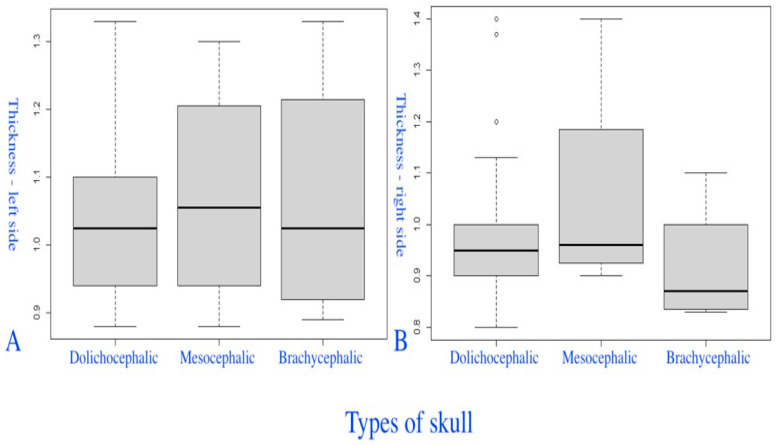
Box-plot diagrams of the distribution of the thickness of the left PM (**A**) and the right PM (**B**) in different types of skull.

**Figure 12 diagnostics-14-01161-f012:**
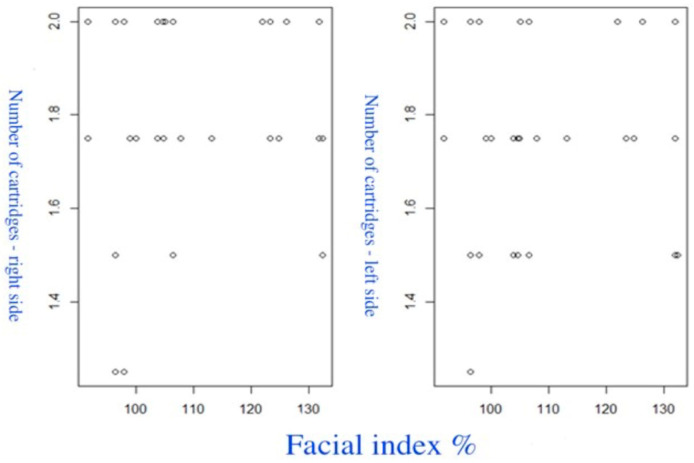
Scatterplot on the relationship between the FI and the number of cartridges.

**Figure 13 diagnostics-14-01161-f013:**
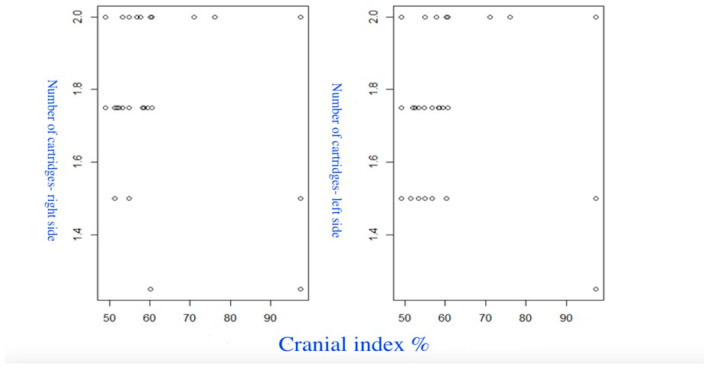
Scatterplots of the relationship between the CI and the cartridges number.

**Figure 14 diagnostics-14-01161-f014:**
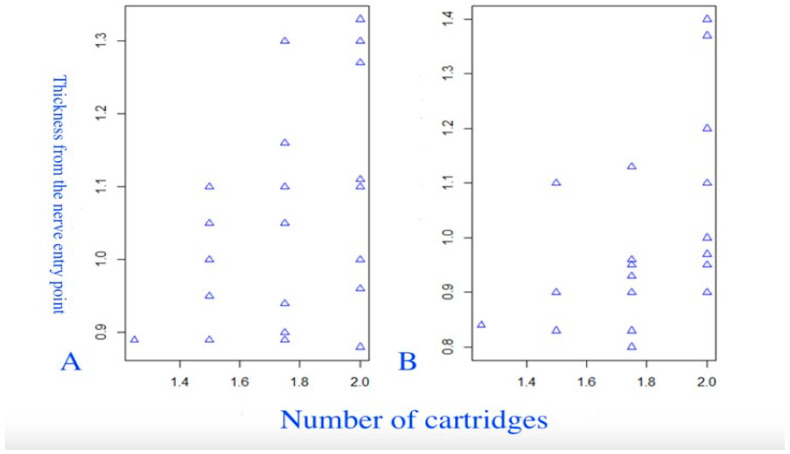
Scatterplots of the relationship between the thickness distance from the nerve entry point and the number of cartridges. (**A**) The left PM. (**B**) The right PM.

**Table 1 diagnostics-14-01161-t001:** Characteristics used to study the volume of the PM.

Indicator	M ±SD	Minimum	Maximum	Me (Q1; Q2)
Age	72.5 ± 12.88	52	102	73 (60.75–81)
FI%	110.65 ± 12.94	91.74	132.35	105.13 (99.29–122.96)
CI%	58.51 ± 13.28	49.04	97.4	58.51 (53.71–68.44)
**PM on the right side (cm)**				
Length	3.6 ± 0.74	2.44	4.5	3.6 (2.92–4.37)
Width	2.12 ± 0.54	1.0	3.5	2.2 (1.99–2.37)
**PM on the left side (cm)**				
Length	3.56 ± 0.85	2.3	5.0	3.6 (2.73–4.0)
Width	2.27 ± 0.42	1.6	3.1	2.17 (2.0–2.57)
**The thickness of the PM from the point of entry of the nerve (cm)**				
Left side	1.05 ± 0.14	0.88	1.33	1.02 (0.94–1.11)
Right side	0.99 ± 0.16	0.80	1.40	0.95 (0.90–1.00)

## Data Availability

The raw data supporting the conclusions of this article will be made available by the authors on request.

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
