# Peer review of "The Pterygomandibular Space: A Volumetric Evaluation Using the Novel A-Silicone Injections Method"

_diagnostics, 2024, doi:10.3390/diagnostics14111161_

Round 1

Reviewer 1 Report

Comments and Suggestions for Authors

The presented study and obtained results have a great practical value for all practitioners who perform mandibular block. The study is an attempt of non-radiological measurement of pterygomandibular space anatomical values. However,  the results presentation must be improved. Figures 2, 5, 8 and 10 show only distribution of scatterplots without any linear relationship between assessed values. Moreover, graphical quality of Figure 2 is not sufficient and must be improved and presented separately for left and right side. There are differences in age ranges between data descripted in the Material and Methods subsection and  data included in the Table 1. Name of the A-Silicone material used in the research must be completed. In the discussion subsection authors did not attempt to explain the obtained results and refer them to other previous studies. I expected more extensive discussion including limitations of the study. I suggest more concise conclusions.

Author Response

Dear Reviewer 1

Thank you for your valuable comments on our paper, and please accept our apologies if our study did not meet your expectations. This time we revised our paper, wishing it would satisfy all the quality standards required by you and the journal. Hopefully, you will enjoy reading and evaluating the revised version of the submitted manuscript.

  1. Regarding Figures 2, 5, 8, and 10, you are right that there is no linear relationship between the assessed values. However, our study mainly aimed to present the distribution of scatterplots.
  2. Based on your suggestion, we improved the quality of Figure 2 (Lines 117-118).
  3. Thank you very much for your attention to the differences in age ranges between the data described in the Material and Methods section and the data included in Table 1. We made all revised our paper accordingly (Line 59).
  4. In the Methods and Materials section, we added details on the injected silicone (S4 Suhy, Bisico®, Bielefeld, Germany) (Line 78). BISICO 2020. S4 suhy. https://bisico.de/en/product/s4-suhy-en/(last access: 27.05.2024).
  5. In our study, we faced a catastrophic lack of similar studies, and it was difficult to compare our results with the data of other authors. In the literature, only two authors wrote about this topic (Rabinovich 2022 and Leven 1987). However, the data published by the authors were vastly limited and did not contain precise data. We can only rely on our results. References included in the list:

(Leven, I.; Increasing the effectiveness of conduction mandibular anesthesia dis. for the job application scientist step. PhD (Candidate of Medical Science), Moscow Medical Dental Institute was named after N. A. Semashko, Moscow, Russia, 1987.)

(Rabinovich, S.; Grin, M.; Omerelli, E.; Velichko, E.; Dashkova, O.; Kheygetyan, A.; Karammayeva, M.; Semo, S. Clinical experience of the use of anatomically guided inferior alveolar nerve blockage in the mandibular teeth treatment. Russian Journal of Operative Surgery and Clinical Anatomy. 2022, 37-43.)

  1. In the Conclusions section, we considered your recommendations.

Reviewer 2 Report

Comments and Suggestions for Authors

Dear Authors,

Congratulations on the job you have done and presented in this manuscript. I believe that your work is very significant to the field and might be of high interest for the general reader, therefore I will recommend publication of your work after some minor modifications. please see the attachment

Comments on the Quality of English Language

Minor editing required

Author Response

Dear Reviewer 2,

Thank you for your valuable comments on our paper, and please accept our apologies if our study did not meet your expectations. This time we revised our paper, wishing it would satisfy all the quality standards required by you and the journal. Hopefully, you will enjoy reading and evaluating the revised version of the submitted manuscript.

  1. We arranged the keywords in alphabetical order (anatomy; anesthesia; dentistry; local anesthesia; mandibular nerve; pterygomandibular space) (Lines 28-30).
  2. The tested null hypothesis is already added to the Introduction section; (H0): there is no correlation between craniometric parameters (the Cranial Index and Izard’s Facial Index), and the volume of PM (length, width, and thickness) on the left and right (Lines 51-54).
  3. An ethical statement dose exists at the end of the Materials and Methods Section and Institutional Review Board Statement Section. The study was conducted in accordance with the Declaration of Helsinki, and approved by the Ethics Committee of Sechenov University, Moscow, Russia 318 (Protocol Number 02-23; January 26, 2023) (Lines 95-97 and 538-540).
  4. We already discussed the limitations of our study (Lines 464-467). However, we improved the discussion section to meet your expectations (Lines 278-298). Also we added some data to the discussion (at lines 291-298).
  5. Conclusions are improved according to your suggestions (Lines 311-314).

Sincerely,

Authors